# In Vitro and Ex Vivo Biofilm-Forming Ability of *Rhinocladiella similis* and *Trichophyton rubrum* Isolated from a Mixed Onychomycosis Case

**DOI:** 10.3390/jof9070696

**Published:** 2023-06-23

**Authors:** Polyana de Souza Costa, Maria Eduarda Basso, Melyssa Negri, Terezinha Inez Estivalet Svidzinski

**Affiliations:** Medical Mycology Laboratory, Department of Clinical Analysis and Biomedicine, State University of Maringá, Maringá 87020-900, Brazil; polyanadesouzacosta4@gmail.com (P.d.S.C.); ra126600@uem.br (M.E.B.); mfnngrassi2@uem.br (M.N.)

**Keywords:** mixed fungal biofilm, black yeast-like, dermatophyte

## Abstract

Infections caused by biofilm-forming agents have important implications for world health. Mixed infections, caused by more than one etiological agent, are also an emerging problem, especially regarding the standardization of effective diagnosis and treatment methods. Cases of mixed onychomycosis (OM) have been reported; however, studies on the microbial interactions between the different fungi in biofilms formed on nails are still scarce. We describe a case of mixed OM caused by the dermatophyte *Trichophyton rubrum* and the black yeast-like fungus *Rhinocladiella similis*. Identical growths of both fungi were observed in more than 50 cultures from different nail samples. Additionally, both species were able to form organized single and mixed biofilms, reinforcing the participation of both fungi in the etiology of this OM case. *R. similis* seemed to grow faster during the process, suggesting that *T. rubrum* benefits from biofilm development when in combination. Moreover, the biofilm of the *Rhinocladiella* isolate exhibited exacerbated production of the extracellular matrix, which was not observed with that of a *Rhinocladiella* reference strain, suggesting that the isolate had natural abilities that were possibly perfected during development in the nail of the patient.

## 1. Introduction

It is estimated the onychomycosis (OM) affects 5.5–11.4% of the world population [1,2]. This fungal infection damages and deforms the nails, causing physical and emotional symptoms and making daily activities difficult [3]. Regarding the pathogenesis of fungal nail infections, although some classical fungal virulence factors such as keratinases and lipases contribute to nail dystrophy [4,5], they have been attributed also to the ability of fungi to organize themselves into biofilm form, with impact on antifungal resistance in onychomycosis [6]. Biofilm is a complex and orderly community that works to maintain its members and keep them nourished and protected from external threats.

Data suggest that mixed OM has been underestimated over the years and that advancing age increases the risk of the presence of more than one etiological agent [7]. The diagnosis of mixed fungal infections has drawn attention, with endless associations, including in biofilm form, described in the literature in recent years. However, the presence of non-dermatophyte molds (NDMs) and other unusual fungal species is still often overlooked in mixed infections and is generally considered environmental contamination [8].

Each fungal association has its own particularity, and fungi tend to adapt and enhance their development when in a community. Here, we describe a mixed OM case involving the most known agent of this disease, the dermatophyte *Trichophyton rubrum*, and an unusual agent, the black yeast-like fungus *Rhinocladiella similis*.

*Rhinocladiella similis* is widely present in environmental substrates such as wood, water, and soil [9]. *Rhinocladiella* spp. are potential agents of human infections such as chromoblastomycosis, which affects the subcutaneous tissue thus causing nodular and verrucous lesions [10,11,12,13]. Furthermore, species of this genus are emerging in patients with nosocomial infections [14], and a pseudo-outbreak has even been reported [15]. Infections involving *Rhinocladiella* spp. require attention, since they have a profile of low sensitivity to echinocandins [16]. However, little is known regarding the pathogenic mechanisms of this fungus during infection. A prospective study of 20,746 samples of skin and nails found that 108 (0.5%) samples were positive for different species of black yeast-like fungi [17]. In addition to this, OM caused by other species of black yeast-like fungi have been reported in the literature [18,19,20,21]. In 2022, a study evaluated the incidence of OM in two different socioeconomic groups and, among the fifty positive samples, seven (14%) involved two etiological agents, and two involved black yeast in co-infections, *Rhinocladiella* spp. with *Candida parapsilosis*, and *Exophiala* spp. with a *Rhodotorula* spp. [22]. It is worth mentioning that *R. similis* was not identified from nail scales in any of these cases. Therefore, the present study reports a mixed OM case involving *T. rubrum* and *R. similis* and aims to describe the behavior of these two fungal agents in single and mixed biofilms in vitro and ex vivo on sterilized healthy human nails.

## 2. Materials and Methods

### 2.1. Patient History and Fungal Samples

An immunocompetent man, 36 years old, had three OM-affected nails on each foot. The patient had OM for 15 years and, during this time, he had tried medicines and homemade recipes. His work involved traveling as a sales representative and in his free time he practiced motocross, wearing closed toe shoes most of the time. For the collection of nail scales, the patient did not take any type of treatment for 3 months, and the scrapings were performed after cleaning the nails with alcohol. The samples were forwarded to the Teaching and Research Laboratory in Clinical Analysis (LEPAC) of the Division of Medical Mycology at the State University of Maringá.

The nail scrapings were submitted to direct mycological examination (DME; 20% potassium hydroxide with Evan’s blue, 3:1 *v*/*v*) and cultured at three points of inoculation in each glass tube containing either Sabouraud dextrose agar (SDA, n = 3; Kasvi, São José dos Pinhais, PR, Brazil) or Mycosel agar (n = 3; BD, BBL™, Les Merciers, Rhone-Alpes, France), totaling 18 inoculations. The cultures were incubated at 25 °C, with daily observation. To confirm mixed infection, culturing was performed on three different days over a period of three months. When consistent growth was observed at the point of inoculation, the fungal colonies were reisolated in new tubes or on glass plates with SDA and presumptively identified by the macromorphological and micromorphological characteristics.

This study was conducted according to the resolution 196/1996 from the National Council of Health (*Conselho Nacional de Saúde*, CNS-MS) under the supervision of the Standing Committee on Ethics in Research Involving Human Beings. The patient provided written and signed consent for publication of the images in Figure 1.

### 2.2. Molecular Identification

The clinical isolates were forwarded to the Microbiological Collections of the Paraná-TAXonline Network at the Federal University of Paraná and deposited under number CMRP5486 (*T. rubrum*) and CMRP5487 (*R. similis*). The isolates were molecularly identified through the polymerase chain reaction (PCR) method amplifying the V9G and LS266 genes from fungal DNA. Sequences obtained were blasted in NCBI Blast (http://www.ncbi.nlm.nih.gov/BLAST, accessed on 15 January 2023), edited in BioEdit version 7.2.5, and a phylogenetic tree was generated using MEGA11 software. A strain of *R. similis* (CMRP3079) from the Microbiological Collections of the Paraná-TAXonline Network was used as a reference (RSR).

### 2.3. The Preparation of Inoculum

The inoculum of each of the two clinical isolates and the reference were prepared from 6-day cultures. The colonies were scraped gently, placed in 0.85% saline, and vortexed vigorously. Suspensions were filtered in a syringe with sterile glass wool to separate hyphae from conidia, and the conidia were counted on a Neubauer chamber (CRAL, Cotia, SP, Brazil). The inoculum concentration of each suspension was adjusted to 1–2 × 10^7^ conidia/mL.

### 2.4. Production of Single and Mixed Fungal Biofilms

#### Formation of Biofilms on Polystyrene Plate

For the single biofilm, in 24-well plates, 200 µL of the standardized inoculum and 1000 µL of an RPMI 1640 medium (Sigma-Aldrich, St. Louis, MO, USA) were added to each well, while for the mixed biofilm, 100 µL of each inoculum and 1000 µL of RPMI 1640 media were added. All cultures were incubated for seven days at 35–37 °C and stirred at 110 rpm.

### 2.5. Ex Vivo Formation of Biofilms on Healthy Nails

Healthy human nail fragments measuring approximately 0.5 × 0.5 cm^2^ were degreased with 99.8% alcohol for 24 h, then rinsed under running water and sterilized via autoclavation. After drying, the fragments were placed on clean slides, with the ventral surface facing up. For the single biofilm, 3 µL of the standardized inoculum was carefully distributed over each nail fragment. For the mixed biofilm, the nails were infected with a suspension prepared at a 1:1 ratio of the isolates and/or reference. The infected nails were kept in a humid chamber and incubated for seven days at 35–37 °C. The growth of all biofilms was evaluated at 24, 96, and 168 h.

### 2.6. Number of Cultivable Cells and Metabolic Activity of the Biofilms

For the quantification of the number of cultivable cells, nail fragments (n = 3/group) were individually transferred to microtubes containing 1 mL of saline and vortexed vigorously for 5 min. The resulting suspensions were serially diluted 1:10 in saline, 10 µL of each dilution was plated on SDA using the dripping technique, and the plates were incubated at 25 °C for 48 h. After this, the colony-forming units (CFUs) per mL were determined.

For evaluation of the mitochondrial metabolic activity, nail fragments (n = 3/group) were transferred to individual microtubes containing 2,3-bis-(2-methoxy-4-nitro-5-sulfophenyl)-2h-tetrazolium-5-carboxanilide solution (XTT; Sigma-Aldrich, St. Louis, MO, USA), incubated at 35–37 °C under stirring at 110 rpm for 3 h, and then read at 492 nm in a microplate spectrophotometer (SpectraMax^®^ Plus 384, Molecular Devices, San Jose, CA, USA).

### 2.7. Scanning Electron Microscopy (SEM)

To visualize the samples in a scanning electron microscope (Shimadzu SS-550, Shimadzu Corporation, Kyoto, Japan), the infected nail fragments were prepared as described in Veiga et al. (2022) [23], including fixation in a 0.1-M cacodylate buffer with 2.5% glutaraldehyde, gradual dehydration in alcohol (50%, 70%, 80%, 90%, 95%, and 100%), critical point drying (BAL-TEC CPD 030), and metallization (BAL-TEC-SCD 050).

### 2.8. Statistical Analysis

Statistical analyses were performed using GraphPad Prism 5 software (San Diego, CA, USA) with two-way analysis of variance (ANOVA) and subsequent subjecting to the Bonferroni multiple comparison test. Data with non-normal distribution were expressed as mean ± standard deviation (SD). *p* values < 0.05 were considered statistically significant. For each test, at least three independent experiments were performed on three different days.

## 3. Results

A 36-year-old man had three infected nails on each foot (Figure 1A,B), with undulations and thickening on the nail surface and detachment of the nail plate. In the DME, abundant fungal structures were observed, with hyaline septate hyphae and some darker stained structures resembling small chains of roundish cells (Figure 1C). Consistent growth occurred at all inoculation points on the seventh day in both culture media (SDA and Mycosel). In each case, two distinct kinds of colony were observed, one white and the other black (Figure 1D). Both colony types were reisolated in SDA and presumptively identified according to macromorphology and micromorphology. The white colony (Figure 1E,G) demonstrated drop-shaped microconidia connected to thin hyphae and hyaline characteristics, while the black colony (Figure 1F,H) had olivaceous hyphae, with the apices forming small bunches full of cylindrical conidia, narrowing towards the base.

Figure 2 presents the behavior of each isolate in single and mixed biofilms, produced on a polystyrene surface with an RPMI medium and on sterilized healthy nail fragments ex vivo. The number of fungal cells recovered from biofilms developed in polystyrene plates (Figure 2A) suggests that the single *T. rubrum* biofilm developed slowly and gradually, while that of *R. similis* had a rapid development, but not gradual, as was also seen with the mixed biofilm. There was, however, no statistically significant difference in the number of cells between the timepoints for each biofilm. For these biofilms, that of *T. rubrum* demonstrated the same behavior in terms of metabolic activity (Figure 2B) as seen with the CFU, that being a slow and gradual increase, while the *R. similis* single biofilm presented a rapid and gradual increase in metabolic activity, with a similar profile seen in the mixed biofilm.

Figure 2C presents the CFU of fungal cells detached from biofilms produced on sterilized healthy nails. Significant increases over time (from 24 to 168 h, *p* < 0.0001) were observed for all biofilms. As observed with the CFU profile, the metabolic activity of all biofilms on the nails presented a significant increase over time (Figure 2D; from 24 to 168 h, *p* < 0.0001).

To better understand the participation of each isolate in the formation of the mixed biofilm, colonies were developed on SDA plates through the dripping technique using fungal cell suspensions recovered from the biofilm. The black yeast-like fungus *R. similis* (darker coloration in Figure 3A and pale gray in Figure 3B) grew more than the dermatophyte *T. rubrum* (white spots in Figure 3A,B). Although the *R. similis* reference (Figure 3B) had a lighter, more grayish color than the *R. similis* isolate from the patient, the growth was more predominant in both cases with the *T. rubrum* clinical isolate. 

SEM micrographs of the biofilms were obtained to enable the assessment and comparison of the behavior of each single and mixed biofilm (Figure 4). The *T. rubrum* single biofilm (Figure 4A–C) presented isolated conidia at 24 and 96 h, and at 168 h some true hyphae could be observed. In 24 h, reproductive cells (conidia) of *R. similis* were adhered on the surface of the nail (Figure 4D), exactly at the place where the inoculum was deposited. In 96 h, the formation of the single biofilms for both the *R. similis* clinical isolate (Figure 4E) and reference (Figure 4K) was similar, with a high degree of filamentation and radial advancement across the surface of the nail; however, the *R. similis* clinical isolate presented an exacerbated production of the extracellular matrix (ECM) and chain conidia, while the conidia of the *R. similis* reference were isolated. In 168 h, the organization of the hyphae and an increase in the biofilm density were visible in both *R. similis* single biofilms (Figure 4F,L), without spreading over the nail surface; this growth remained limited to the site of the previous observation.

In the mixed biofilm of the clinical isolates, at 24 and 96 h, there was the development of true hyphae, and at 168 h, a biofilm with a predominance of hyphae and a small amount of reproductive cells was observed, while in the mixed biofilm with the reference, the true hyphae were only visualized at 168 h.

It is important to highlight the differentiation that was observed at 96 h of biofilm incubation between the *R. similis* clinical isolate and the reference strain, with the former producing a large amount of MEC, which was focused from the center to the borders of the biofilm growth, presenting larger and denser development, such as that presented in Figure 5.

## 4. Discussion

The presence of more than one agent responsible for the etiology of a case of OM is still not well accepted in clinical routine. Since Koch’s postulates, which were created in the 19th century, the orientation has been to value “pure cultures” only, with the isolation of a single microorganism in each infectious process. However, nowadays, this paradigm should be revised, as it has been proven that microbial infections can be caused by one or more etiological agents at the same time [24].

Mixed infections have been reported in cases of OM [7,21,25,26,27]. However, there is still no consensus within the field regarding a standardized method of diagnosis and, thus, some important discussions have been raised. The use of molecular methods is widely indicated in the diagnosis of OM, especially in mixed infections; however, these methods are prone to errors, such as the amplification of contaminants. Some methodologies, such as histopathological examination, have contributed to the confirmation of polymicrobial infections by enabling the observation of different morphologies in the infected nail tissue. Culture is the reference method in the diagnosis of OM, although it is also susceptible to errors. Stefanato and Verdolini (2009), however, highlighted the importance of Grocott and periodic acid-Schiff (PAS) stains in the identification of mixed OM, since opportunistic fungi, such as *Scopulariopsis brevicaulis*, can mask the growth of other agents in culture, preventing their isolation and further morphological identification [26]. Furthermore, the development of environmental fungi or microbiota generates difficulties in the interpretation of OM etiology. One recommendation to work around this problem is to culture via repeat sampling to obtain reproducible growth. This procedure is an important criterion for confirming molecular test results [7,27]. The rigorous methodology used in this study, with more than 50 cultures identical to those shown in Figure 1D, assures the effective participation of both *T. rubrum* and *R. similis* in the etiology of this infection. The correct diagnosis of mixed OM is important, as the treatment may need to be significantly longer and more complex compared to that of OM caused by a dermatophyte alone [28].

Currently, the pathogenesis of OM has been attributed to fungi organized in biofilm form, and thus, in the current study, single and mixed biofilms were artificially produced with the two fungal isolates on two surfaces: a polystyrene plate and sterilized healthy nails. The ex vivo experiments, without the addition of other nutrient sources, are more representative of how pathogenesis occurs during infection.

Although *T. rubrum* is a dermatophyte, in addition to being the most commonly isolated agent in OM, it seems that its main specialty is not its ability to produce a single biofilm [29]. Information on *Rhinocladiella* spp. infections is lacking, but some species of this genus have been classified as moderate and strong biofilm producers [30].

In the current study, one of the difficulties encountered was regarding the methodology’s ability to individually quantify the fungi detached from the mixed biofilms, in view of the unavailability of differential selective culture media. However, it was possible to estimate that the CFU recovered from the *T. rubrum* single biofilm in polystyrene plates increased slowly and gradually over time, differently to that of the single biofilm of the *R. similis* isolate which, although it did not increase over time, had a higher CFU number at all timepoints. The same trend was detected in the single biofilms produced on the nail: slow development of the *T. rubrum* single biofilm, while that of *R. similis* showed a rapid and gradual development. Regarding the mixed biofilm of the clinical isolates, the CFU counts were high and comparable to those of the *R. similis* single biofilm. Furthermore, the CFU counts were slightly higher on the polystyrene surface in general for all biofilms, probably owing to the availability of nutrients in the RPMI culture medium. On the other hand, it was the mixed biofilm developed on the nails that provided the most valuable information. We show for the first time that *R. similis* is not only able to grow using the nail as the sole nutritional source, but it is also efficient in forming very well-organized single and mixed biofilms when combined with *T. rubrum*.

The metabolic activity of the *T. rubrum* single biofilm on polystyrene plates presented a significant increase up to 96 h with a subtle decrease at 168 h, a profile similar to that already described [31]. According to the authors, this behavior is related to the stationary phase of development of this species. The single biofilm of the *R. similis* clinical isolate in the polystyrene plates, meanwhile, showed greater metabolic activity than its partner, *T. rubrum*, which increased up to the end timepoint of 168 h. The metabolic activity was significantly higher for the *T. rubrum* single biofilm and the mixed biofilm of the clinical isolates in the nails compared to those of the polystyrene plates, showing a three-fold and at least a two-fold increase, respectively. This, once again, highlights the strong interaction of both fungi with the nail’s natural substrate.

To confirm the participation of each species in the mixed biofilm, the growth of fungal cells recovered from the mixed biofilms was qualitatively visualized, through which a greater presence of dark fungus growth, attributable to the *R. similis* isolate, was observed (Figure 3), compatible with quantitative data (Figure 2). These data suggest that *T. rubrum*, which is not very efficient in producing a single biofilm, seems to have benefited from the abilities of *R. similis* in the mixed biofilm. It is possible to presume that *T. rubrum* has contributed with other competences, such as the production of enzymes and proteins [32]. Although there is no proof, it is possible that *R. similis,* like *Fusarium* spp. [33], is another non-dermatophyte fungus capable of keratin degradation. This assumption is supported since *R. similis* isolates matured with true hyphae and produced biofilms, using the human nail as a unique nutritional source. Black yeast-like fungi may play an important role in microbial interactions with other fungi, as a case with mixed OM involving a *C. parapsilosis* and another black yeast-like fungus *Exophiala dermatitidis* was recently described, with similar results for single and mixed biofilms as those of the present study [21].

SEM micrographs of the biofilms produced on the nails ex vivo allowed us to confirm the results described above (Figure 4). Micrographs of *T. rubrum* showed that only isolated conidia adhered to the surface of the nails up to 96 h, and at 168 h the formation of a few true hyphae could be observed along the entire extension of preparation. The *R. similis* biofilm exhibited a rapid production of conidia, demonstrating agility in development (growing and adapting to the natural substrate). Furthermore, the single biofilm of the *R. similis* clinical isolate, but not that of the *R. similis* reference, showed exacerbated production of what appeared to be an extremely dense ECM, which radiated from the center to the edge of the produced biofilm. The biofilm produced by the *R. similis* clinical isolate and reference independently suggest that the ability to grow and adapt to a new substrate is an intrinsic ability of the species. Moreover, as the *R. similis* clinical isolate was more efficient in the production of ECM (Figure 5—complementary material), it suggests that this natural capacity was stimulated and improved during the natural development on the patient’s nail. This observation reinforces the participation of this black yeast-like fungus in the etiology of this OM case. The SEM micrographs of the mixed biofilm of the clinical isolates at 24 h showed conidia-forming clusters with the presence of what appeared to be ECM, true hyphae at 96 h, and many reproductive cells present, and at 168 h, the nail was covered by true hyphae forming tunnels. For the mixed biofilm with *T. rubrum* and the *R. similis* reference, conidia-forming chains were present at 24 h, at 96 h these conidia underwent proliferation, and at 168 true hyphae predominated.

Interactions between individual agents within a mixed biofilm can vary as synergistic, neutral, or antagonistic, depending on the species involved. While significant knowledge exists regarding yeasts such as *Candida* spp., the understanding of filamentous fungi is limited.

The results of the present study suggest a synergistic interaction between *T. rubrum* and *R. similis*, with the sharing of competences between the fungi involved. We conclude that the black yeast-like fungus was able to form a dense and organized biofilm, adapting easily to the nail and using it as its only source of nutrients. More robust methodologies, such as genomic and metabolomic analyses, which are important tools, are needed to better understand the interaction between fungi in mixed infections, particularly with regard to the metabolism and gene expression of biofilms [34,35]. Studies of microbial interactions within a biofilm are extremely important, such as the roles of quorum-sensing molecules, which operate as modulators in the communication between the individuals involved, controlling the formation and architecture of the biofilm [36].

## Figures and Tables

**Figure 1 jof-09-00696-f001:**
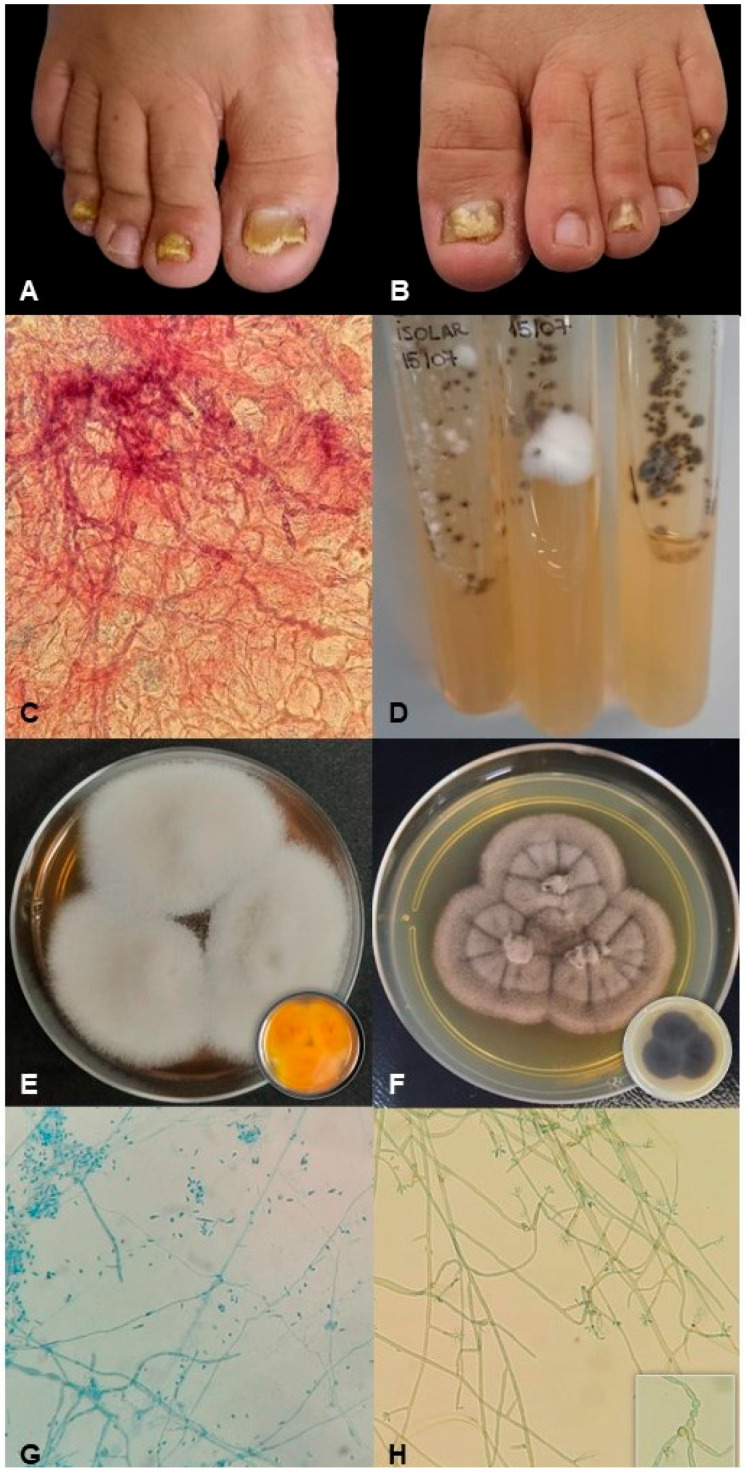
Clinical presentation and laboratory characteristics of a case of onychomycosis. (**A**) Clinical right and (**B**) left toenail aspects; (**C**) fungal structures observed via direct mycological examination; (**D**) nail scrapings cultured at nine points in Sabouraud dextrose agar showing mixed growth after seven days of incubation at 25 °C; (**E**) macromorphology from the top (and bottom, inset) of *Trichophyton rubrum* and (**F**) *Rhinocladiella similis* colonies; and (**G**) micromorphological aspects of *T. rubrum* and (**H**) *R. similis*.

**Figure 2 jof-09-00696-f002:**
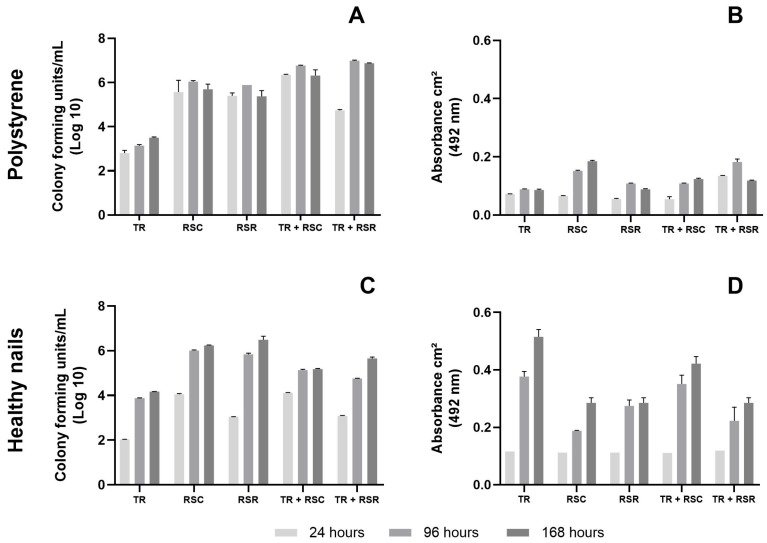
Evaluation of single and mixed biofilms produced (**A**,**B**) in 24-well polystyrene plates with RPMI and (**C**,**D**) on sterilized healthy human nails fragments without other nutrients source. (**A**,**C**) Quantification of viable fungal cells recovered from biofilms in terms of colony-forming units per mL (CFU/mL). (**B**,**D**) Evaluation of metabolic activity using the XTT assay and shown in terms of absorbance at 492 nm. TR, *Trichophyton rubrum* (clinical isolate); RSC, *Rhinocladiella similis* (clinical isolate); RSR, *Rhinocladiella similis* reference.

**Figure 3 jof-09-00696-f003:**
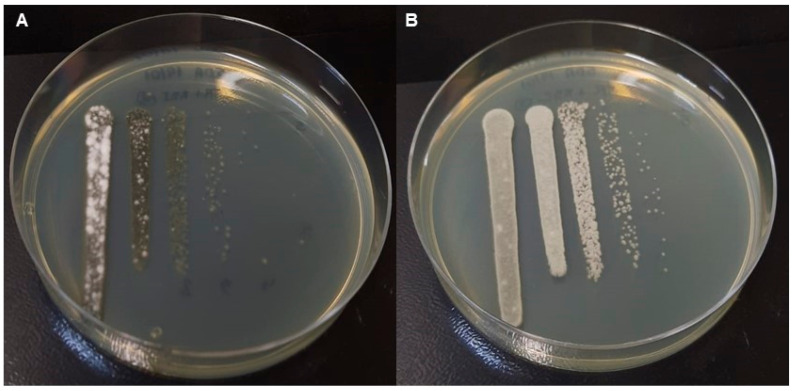
The dripping technique was performed with cells recovered from a mixed biofilm artificially produced on sterilized healthy nail fragments, which were photographed after 72 h of incubation at 25 °C. (**A**) *Trichophyton rubrum* and *Rhinocladiella similis* clinical isolates, and (**B**) the *T. rubrum* clinical isolate and *R. similis* reference.

**Figure 4 jof-09-00696-f004:**
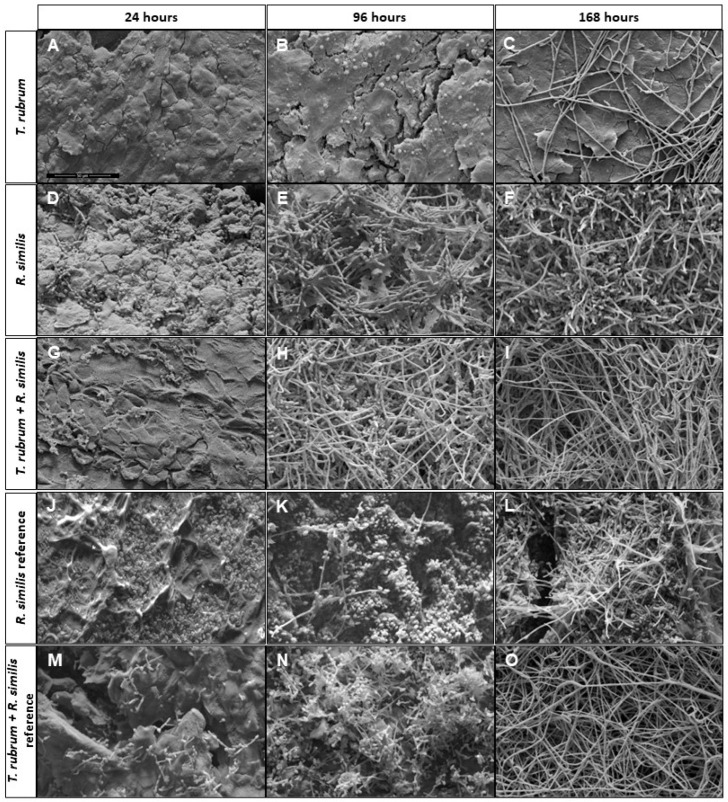
Scanning electron microscopy microphotographs showing the architecture of the single biofilms produced by the *Trichophyton rubrum* (**A**–**C**), *Rhinocladiella similis* clinical isolates (**D**–**F**), and the *R. similis* reference (**J**–**L**) on sterilized healthy human nails. The mixed with the clinical isolates (**G**–**I**) and, the *T. rubrum* clinical isolates and the *R. similis* reference strain (**M**–**O**). SEM images at 2000× magnification.

**Figure 5 jof-09-00696-f005:**
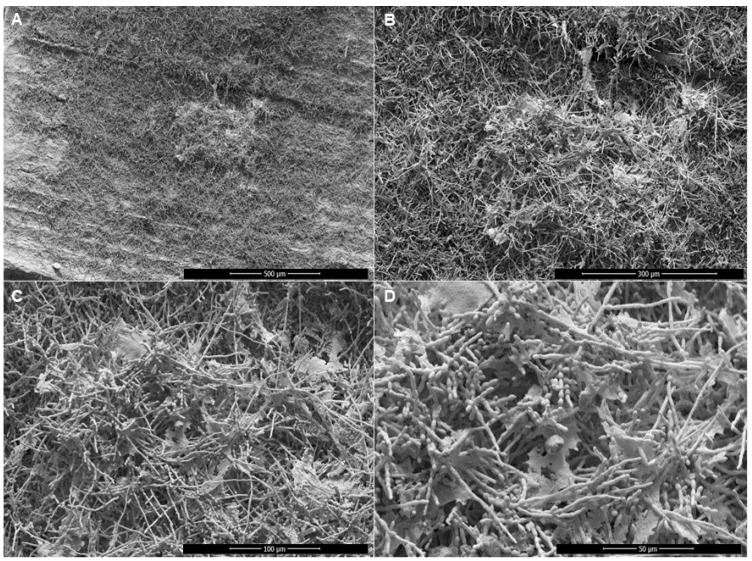
Scanning electron microscopy microphotographs showing the exacerbated production of matrix in biofilm by *Rhinocladiella similis* clinical (CMRP5487) on healthy and sterilized human nails. (**A**,**B**) Images demonstrating the presence of a dense matrix concentrated in the center of the fungal growth; (**C**) abundant presence of conidia arranged in a chain and organized hyphae in development (**D**) with the existence of points of the highly dense extracellular matrix. Microphotographs captured at 200, 500, 1000, and 2000× magnification.

## Data Availability

Not applicable.

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
