# Peer review of "In Vitro and Ex Vivo Biofilm-Forming Ability of Rhinocladiella similis and Trichophyton rubrum Isolated from a Mixed Onychomycosis Case"

_jof, 2023, doi:10.3390/jof9070696_

Round 1

Reviewer 1 Report

Introduction:

Line 41. put the complete name at the beginning of a sentence

line 51: which is the conclusion about the comparison of the two socioeconomical groups? is this relevant? 

line 54. Rhinocladiella similis could be included  in the isolates identified as Rhinocladiella spp if the complete identification had been performed completely to species in the study presented above?

Materials and methods:

Were the sequences of the isolated fungi deposited in Genbank?  

Author Response

Dear reviewer,

We would like to thank the reviewers for careful and thorough reading of this manuscript and for the thoughtful comments and constructive suggestions, which help to improve the quality of this manuscript.

In the manuscript, all changes are evidenced with the “Track Changes” function.

Our response follows (the reviewer’s comments are in italics).

Comments and suggestions from the reviewers to authors:

Reviewer #1:

Introduction:

Line 41. put the complete name at the beginning of a sentence

Author's response: Suggestion accepted.

line 51: which is the conclusion about the comparison of the two socioeconomical groups? is this relevant?

Author's response: The article is very interesting and brings an important look at homeless people, however, regarding mixed infection rates, the data are very similar, as in housed people 3 cases were identified (Aspergillus spp. + Cladosporium spp., Candida parapsilosis + Rhinocladiella spp. and Trichophyton violaceum + Trichophyton schoenieinii) while in homeless, 5 cases (Scytalidium spp. + Onychocolac canadensis, Rhodotorula spp. + Candida parapsilosis and two of Trichophyton Scholenieinii + Aspergilllus spp.). So we thought it wouldn't be exactly as relevant to differentiate in our introduction.

line 54. Rhinocladiella similis could be included in the isolates identified as Rhinocladiella spp if the complete identification had been performed completely to species in the study presented above?

Author's response: Yes, it is possible since in this case of onychomycosis the species R. similis has been identified by molecular tools.

Materials and methods:

Were the sequences of the isolated fungi deposited in Genbank?

Author's response: Not yet, but they are in the process together with the Microbiological Collections of the Paraná-TAXonline where the microorganismos are being deposited.

Kind regards,

Dra. Terezinha Svidzinski.

Reviewer 2 Report

This is an interesting study of fungi biofilm. The authors isolated two fungi, Trichophyton rubrum and Rhinocladiella similis, from the nail sample of onychomycosis (OM) patient. The biofilm formation ability of clinical isolated strain of R. similis is better than reference strain. The authors shall compare the differences between the clinical isolated strain and the reference strain of Rhinocladiella similis. 1. drip technique (Line 127) or dripping technique (Line 195, 202)? 2. Figure 1: ”D” is not labeled in the figure. 3. Figure 2: the figure legend should be revised. 4. Figure 5 shall be added in the text, not in the complementary material. 5. Line 49: 108/20746 shall be about 0.5%.

This is a well-written manuscript that only needs to undergo a few minor changes.

Author Response

Dear reviewer,

We would like to thank the reviewers for careful and thorough reading of this manuscript and for the thoughtful comments and constructive suggestions, which help to improve the quality of this manuscript.

In the manuscript, all changes are evidenced with the “Track Changes” function.

Our response follows (the reviewer’s comments are in italics).

Comments and suggestions from the reviewers to authors:

Reviewer #2:

This is an interesting study of fungi biofilm. The authors isolated two fungi, Trichophyton rubrum and Rhinocladiella similis, from the nail sample of onychomycosis (OM) patient. The biofilm formation ability of clinical isolated strain of R. similis is better than reference strain. The authors shall compare the differences between the clinical isolated strain and the reference strain of Rhinocladiella similis.

  1. drip technique (Line 127) or dripping technique (Line 195, 202)?

Author's response: Suggestion accepted.

  1. Figure 1: ”D” is not labeled in the figure.

Author's response: Suggestion accepted.

  1. Figure 2: the figure legend should be revised.

Author's response: Suggestion accepted, the legend has been improved.

  1. Figure 5 shall be added in the text, not in the complementary material.

Author's response: Suggestion accepted.

  1. Line 49: 108/20746 shall be about 0.5%.

Author's response: Suggestion accepted.

Kind regards,

Dra. Terezinha Svidzinski.

Round 2

Reviewer 2 Report

I have no further comments.

Author Response

Ok, thank you.